# Identifying Access Barriers to PrEP Among Cisgender Black/African American Women in the United States: A Systematic Review of the Literature

**DOI:** 10.3390/healthcare13010086

**Published:** 2025-01-06

**Authors:** JoAnna Boudreaux, Cristobal Mario Valdebenito, Latrice C. Pichon

**Affiliations:** Division of Social and Behavioral Sciences, School of Public Health, University of Memphis, Memphis, TN 38152, USA; jbdreaux@memphis.edu (J.B.); cristobal.v@memphis.edu (C.M.V.)

**Keywords:** pre-exposure prophylaxis, Black women, HIV

## Abstract

Background/Objectives: Cisgender Black women in the U.S. face disproportionately high HIV rates due to systemic inequities rooted in institutional racism, not individual behaviors. These disparities are particularly severe in the southern U.S., driven by limited access to healthcare, economic instability, and unsafe social environments. Despite its proven effectiveness, PrEP remains significantly underutilized in this population. Methods: This systematic review followed PRISMA guidelines to identify and select relevant studies and used the CASP checklist to appraise the quality of the selected qualitative studies. The review focuses on individual and systemic barriers to PrEP access for cisgender Black women, aiming to guide equitable health interventions and improve HIV prevention efforts. Results: Key barriers include limited PrEP awareness, medical mistrust, and stigma. Financial, structural, and social determinants also hinder access. Facilitators, such as PrEP education, social normalization, trust building, and affordability, were identified as critical to improving uptake. Conclusions: The findings emphasize the need for culturally tailored strategies that build trust, provide education, and empower cisgender Black women to overcome barriers to PrEP access.

## 1. Introduction

According to the Centers for Disease Control and Prevention (CDC), Black individuals in the United States continue to be disproportionally affected by HIV [1]. Cisgender Black women, specifically, appear to be at increased vulnerability compared to their White and Hispanic counterparts [1]. Overall, cisgender women account for 19% of new HIV diagnoses, but cisgender Black women account for half of those cases, while only representing 12.9% of reproductive women in the United States [2]. These disparities are even more pronounced in the southern region of the United States, where half of new cases are reported despite the south accounting for only 37% of the United States population [1]. Most new HIV cases (91%) among cisgender Black women can be attributed to heterosexual contact [1].

The overall disparity in HIV acquisition rates amongst the Black/African American population is deeply rooted in institutional racism, which has created conditions—such as poverty, segregation, and mass incarceration—that increase vulnerability to syndemic health threats [3]. Unlike what might be assumed, this disparity is not driven by a higher prevalence of individual risk behaviors [3,4]. Research shows that Black women, for example, tend to report fewer sexual partners and higher rates of condom use than women from other racial groups [5,6]. However, as Aral et al. (2008) [4] point out, Black women can still face high STI rates despite low-risk behaviors, due to elevated HIV acquisition rates within their social networks. The disproportionate rates of STIs may be connected to the sex ratio imbalance between Black men and women, influenced by higher mortality rates and mass incarceration among Black men. This imbalance exposes Black women to sexual networks with higher HIV prevalence and can shift relationship power dynamics [4,5,7].

Furthermore, centuries of institutional racism and discrimination have resulted in deeply embedded social inequities in environments where Black people live, work, worship, and play [5]. These inequities manifest in limited access to affordable housing, quality education, safety, nutritious food, and healthcare—collectively referred to as social determinants of health [8]. As a result, cisgender Black women often have limited access to healthcare education and may lack awareness of their vulnerability to HIV and other STIs. Addressing these racial and geographic disparities has become a priority in the United States Plan for Ending the HIV Epidemic (EHE), which aims to increase access to preventive interventions like pre-exposure prophylaxis (PrEP) for those most affected. The goal of the EHE is to increase the estimated percentage of people with indications for PrEP classified as having been prescribed PrEP to at least 50% by 2025 and remain at 50% by 2030 [9].

CDC estimates that of the 1.2 million people in the United States who could benefit from PrEP, only 30% were prescribed PrEP in 2021 [10]. Notably, approximately 400,000 cisgender Black women are among those who would benefit from PrEP [10]. PrEP is a daily medication regimen that prevents HIV transmission by combining two antiretroviral drugs, tenofovir and emtricitabine. After several successful randomized trials demonstrating its effectiveness in reducing HIV risk for individuals at substantial risk—including those with HIV-positive partners, people who inject drugs, and others with heightened exposure risk—PrEP was approved by the FDA in 2012 [11,12]. Following this approval, the CDC released clinical guidelines in 2014 recommending PrEP for HIV prevention. Since then, the number of people taking PrEP to prevent HIV has significantly increased. In 2021, approximately 30% of the 1.2 million individuals who could benefit from PrEP were prescribed it, compared to just 13% in 2017 [13]. However, despite this progress, Black and Hispanic/Latino people—who represent the majority of those who could benefit from PrEP—remain underrepresented among those prescribed it.

Over the years, there has been significant research and advocacy focused on improving access to and uptake of PrEP, primarily targeting men who have sex with men (MSM) [14,15,16]. MSM are recognized as a key population at high risk for HIV transmission. However, it is crucial to acknowledge that women, especially those from marginalized communities such as cisgender Black women, also face increased vulnerability to HIV acquisition [17]. This population’s risk is compounded by structural violence, including poverty, underemployment, unstable housing, and community violence, which significantly inhibits their capacity to initiate and prioritize PrEP [9,11,12,17]. MSM and women have distinctly different risk profiles for acquiring HIV, highlighting the importance of understanding the specific barriers encountered by women. By recognizing and addressing these barriers, including the societal-level challenges such as the lack of targeted PrEP awareness campaigns, we can ensure equitable access to PrEP and effectively reduce HIV transmission rates among women.

We conducted a systematic literature review on qualitative studies that aimed to explore and understand barriers to PrEP encountered by cisgender Black women in the United States. Systematic reviews offer a comprehensive and structured synthesis of existing research, providing a clear understanding of the state of knowledge on specific issues. Examining barriers to PrEP access among cisgender Black women is critical, as they face distinct HIV risk profiles and unique challenges. This review consolidates insights into their experiences, identifies gaps in current interventions, and supports the development of tailored, equitable, and effective strategies to improve health outcomes for this underserved group. By examining existing studies, we aim to identify specific challenges—such as structural inequities, cultural factors, and healthcare access issues—and inform future research and the creation of interventions designed to reduce HIV transmission rates in this marginalized population.

Data gathered from qualitative studies provide deeper and more detailed information exploring people’s perspectives and lived experiences. While quantitative studies are invaluable for measuring prevalence outcomes, they often fail to capture the nuanced reasons behind health disparities, particularly in marginalized populations. Information gathered from qualitative data can help identify gaps in knowledge and inform future research directions. Most importantly, qualitative data can provide a theoretical underpinning for the development of health interventions. Guided by an intersectional approach, this review examines how race and gender-based discrimination intersect to create specific barriers for Black cisgender women in accessing PrEP. Our review of relevant qualitative studies aimed to answer the following research questions:What are individual barriers to PrEP access for Black women in the United States?What are systematic barriers to PrEP access for Black women in the United States?How can PrEP access be better facilitated for Black women in the United States?

Our focus on qualitative studies is meant to complement quantitative research and discover overlooked aspects of health behavior that can only be captured in personal accounts. Moreover, the focus on Black women in this systematic review stems from their unique and disproportionate burden of HIV in the United States [1]. By centering the perspectives of Black women, this review aims to provide nuanced insights into the specific barriers they encounter as well as provide actionable recommendations to improve healthcare access and HIV prevention efforts tailored to their needs.

## 2. Materials and Methods

We followed the Preferred Reporting Items for Systematic Reviews and Meta-Analysis (PRISMA-P 2015) guidelines [18]. The main objective is to better understand the qualitative literature exploring barriers and facilitators to PrEP uptake among Black cisgender women in the United States. Thus, the population of interest was Black cisgender women living in the United States with negative or unknown HIV status. The age range of the women in the studies was 18 and over.

J.B. worked in consultation with a health sciences librarian (IS) at the University of Memphis to develop a Boolean search strategy using various combinations of relevant MeSH terms and keywords: “African American women” OR “Black women”, AND “pre-exposure prophylaxis”, OR “preexposure prophylaxis” OR “HIV prevention”, OR “Truvada”, OR “treatment as prevention”. During our preliminary exploratory searches, we included “South” and “Deep South” in our keywords since we originally intended to focus on articles discussing PrEP access among cisgender Black women in the Southern geographic area of the United States. However, because we had such limited results, we decided to omit these terms from our search to expand to Black cisgender women across the entirety of the United States. We then conducted searches across five electronic databases: Biomed Central Public Health (BMC), CINAHL, Scopus, PubMed, and Google Scholar.

Since FDA approval of PrEP did not occur until 2012, we limited our search criteria to between January 2012 and October 2024. We selected (1) studies written in English; (2) studies published in peer-reviewed academic journals; (3) studies with full texts available; (4) qualitative studies or mixed-method studies in which we focused on the qualitative component; and (5) studies with the primary aim of exploring access barriers among Black cisgender women in the United States.

Data extraction was performed by J.B. The initial search results from the five online databases yielded 1218 potential articles. These results are from PubMed (*n* = 149), Scopus (*n* = 40), CINAHL (*n* = 64), BMC (*n* = 260), and Google Scholar (*n* = 705). We organized our findings in a Microsoft Excel sheet. Duplications were removed (*n* = 95). Articles were removed if their titles or abstracts indicated that they were review papers, conference abstracts, or quantitative studies, were not focused on women living in the United States, included transgender women, adolescents, women who were substance users, or women in the criminal justice system, or were otherwise irrelevant (*n* = 1017). We omitted studies that centered on mass incarceration and HIV prevention efforts, including women within the prison system. Women in the prison system are disproportionately affected by HIV due to several intersecting dynamics that render them vulnerable to both HIV and incarceration [19,20]. We also omitted studies focused on adolescents (defined as being aged between 10 and 17 years old). Adolescents have multiple complex risk factors that are bound to be different than those of adult women [21]. Finally, we did not include studies with transgender women due to unique characteristics and risk factors compared to cisgender women [22]. Based on this screening, 1070 were removed.

J.B. independently screened the 53 remaining titles and abstracts. When relevancy was unclear, the full text of the article was obtained and further reviewed. Several studies were excluded because their titles and abstracts did not clearly indicate their relevance to the focus on Black cisgender women and PrEP uptake. These included studies on healthcare workers, dissertations, conference abstracts, review articles, and other topics outside the scope of this review. We have clarified these exclusions to ensure transparency in the selection process. J.B. then collaborated with C.M.V to create a structured charting table that included study information, population of interest, sample size, study methodology, applied theories, and findings. Based on this table, J.B. made the final selection of 13 articles, and C.M.V. confirmed this decision.

After reaching consensus on the 13 selected articles, J.B. and C.M.V independently analyzed and coded each article to mitigate the risk of bias. Although no significant discrepancies arose, any minor differences were resolved informally through an iterative discussion process. We systematically extracted data on study design, participant characteristics, interventions, and outcomes using a standardized coding framework. The primary outcomes focused on barriers and facilitators to PrEP uptake among Black cisgender women, while secondary outcomes included knowledge, perceptions of PrEP, and sociocultural factors influencing its uptake. Data were collected on key variables such as participant characteristics (age, geographic location, HIV status), intervention details (e.g., PrEP counseling), and study characteristics (e.g., sample size, methodology). All studies provided complete data. No automation tools were used during data collection. All extracted data were cross-checked by J.B. and C.M.V. to ensure accuracy and reliability.

In accordance with PRISMA guidelines, a PRISMA flow diagram was created to outline the study selection process. The flow diagram follows the standardized template provided by PRISMA and adheres to the PRISMA checklist to ensure transparency and consistency in reporting the inclusion and exclusion criteria for studies in this systematic review. Although registration of the review protocol was not completed, all review processes were conducted transparently, following established systematic review methodologies (see Figure 1).

The final thirteen articles were further analyzed using the Critical Appraisal Skills Programme (CASP) [23] checklist. This checklist was employed to assess the quality, rigor, and risk of bias in each individual study, with a particular focus on issues of generalizability. See Table 1. 

While all studies scored high across various criteria, several noted limitations regarding geographic constraints, as detailed in Table 1. While these limitations are relevant to individual studies, we believe they do not pose significant constraints on our systematic review. Although each individual study may have focused on a specific region, when considered together, they provide a comprehensive and holistic view of the barriers and facilitators to PrEP uptake among Black cisgender women. This geographic diversity enriches our understanding of how regional differences, as well as shared systemic challenges, influence access to PrEP and HIV prevention efforts across the United States.

The included studies span a variety of regions, reflecting both urban and rural settings and offering insights into the geographic context of PrEP implementation. Several studies were conducted in major metropolitan areas such as New York, NY (e.g., Bond [5], D’Angelo [26], Park [28]), Chicago, IL (e.g., Pyra [29], Hirschhorn [34]), and Atlanta, GA (e.g., Auerbach [25], Chandler [32]). Others focused on the southeastern United States, including Jackson, MS (e.g., Arnold [24], Willie [35]), and broader regions such as Chapel Hill, NC, and surrounding states (e.g., Troutman [31], Randolph [30]). Additionally, some studies included multiple sites, such as Auerbach’s work [25], which spanned New York, Dallas, Atlanta, Chicago, and New Orleans.

Given the qualitative nature of this review, no formal meta-analysis was performed. Instead, thematic synthesis was utilized to identify common barriers and facilitators to PrEP uptake, with studies grouped by thematic findings. The risk of bias was assessed for each study, with an overall categorization of low to moderate risk. Factors contributing to this assessment included the clarity of study objectives, appropriateness of the qualitative methodology, recruitment strategies, data collection methods, and the rigor of data analysis. No sensitivity analyses or statistical syntheses were conducted, as these are not applicable in qualitative research. The overall certainty in the findings is moderate, given the diversity in location contexts.

## 3. Results

A total of 13 articles were identified. All studies focused on the majority of Black cisgender women considered at risk for HIV. While all thirteen studies explored barriers to PrEP, not all studies investigated possible facilitators. Specifically, nine studies (*n* = 9) explored both barriers and possible facilitators to PrEP uptake and adherence [5,24,25,26,27,28,29,30,31]. Four studies (*n* = 4) centered solely on barriers to women’s willingness to take PrEP and adhere to the medication long term [32,33,34,35].

Some studies focused on a certain subset of Black cisgender women. Bond et al. (2022) [5] focused on “young” Black women, defined as between the ages of 18 and 25. Nydegger et al. (2021) [27] considered how the women of their study were affected by interpersonal violence and structural poverty. We included these articles because they met the overall search criteria, having the same characteristics of our priority demographic in that they were adult cisgender Black women with negative or unknown HIV status who had never taken PrEP.

In most studies, participants were recruited through healthcare clinics, community-based organizations, HIV testing sites, and homeless shelters. The only exception is Troutman et al. (2021), in which participants living in the Southeastern United States were invited to take an online survey and subsequently recruited to participate in an online focus group [31]. Six studies (*n* = 6) were comprised strictly of semi-structured qualitative interviews [24,26,27,28,32,35]. Four studies (*n* = 4) utilized qualitative focus groups [5,25,30,33]. Three studies (*n* = 3) employed a mixed-methods approach, of which we only reviewed the results from the qualitative component [29,31,34].

Of the thirteen studies (*n* = 13), the majority (*n* = 12) indicated that thematic analysis was used to analyze the data. Bond et al. (2022) [5] used the frameworks of intersectionality and the theory of triadic influence (TTI) to guide their analysis.

### 3.1. Barriers to PrEP

After assessing each article for risk of bias, we systematically identified and organized the main themes from the literature, starting with the barriers to PrEP uptake.

Table 2 summarizes the key barriers to PrEP access and uptake identified in the studies reviewed. For each barrier, we provide the number of studies that addressed it, offering a concise overview of the themes and their prevalence across the literature.

#### 3.1.1. Low Levels of PrEP Knowledge

Twelve articles (*n* = 12) identified low levels of PrEP knowledge as a primary barrier [5,24,25,26,27,28,29,30,32,33,34,35]. Many women reported discovering PrEP for the first time during research interviews, expressing surprise that their healthcare providers had not previously informed them about PrEP [29,31,33]. Notably, while participants in Troutman et al. (2021) [31] were aware of PrEP, none had heard about it from their healthcare providers. Some women took the initiative to research PrEP themselves and presented their findings to healthcare professionals, only to encounter discouragement from those providers [28,29]. Moreover, misunderstandings regarding PrEP are common, with many women believing it is exclusively for gay men, a perception likely fueled by targeted advertising and media portrayals [28,29,33].

#### 3.1.2. PrEP Stigma

PrEP stigma emerged as a significant barrier, cited in eleven articles (*n* = 11) [5,24,25,26,27,28,31,32,33,34,35]. Women reported fears of judgment from parents, friends, community members, and partners, expressing concern that using PrEP might lead to assumptions about their sexual behavior or health status. PrEP stigma was particularly pronounced in romantic relationships, where women worried that their partners might reject them for using PrEP, which could raise suspicions of infidelity. Some participants described the difficulty of discussing PrEP with partners, noting that it could be a sensitive topic that may lead to conflict [5,26,27,31]. For example, one woman in Nydegger et al. (2021) [27] mentioned a partner who explicitly discouraged her from taking PrEP, demonstrating the challenges of navigating PrEP stigma in intimate relationships.

#### 3.1.3. Concerns About PrEP Side Effects

Concerns about PrEP side effects were highlighted in nine articles (*n* = 9) [5,24,25,26,28,29,32,34,35]. Women expressed apprehensions regarding the potential impact of PrEP on their overall health, fertility, and internal organs, as well as the risk of harm to unborn babies. Questions arose about possible interactions with other medications and the exacerbation of existing medical conditions. Some participants even voiced concerns that taking PrEP might weaken their immune systems and increase their susceptibility to HIV [5,28]. These concerns underscore the need for effective communication from healthcare providers about the safety and benefits of PrEP.

#### 3.1.4. Structural Poverty and Social Determinants

Structural poverty and social determinants were discussed in seven articles (*n* = 7) [24,25,26,28,30,32,35]. Participants reported that experiencing structural poverty created numerous barriers to accessing HIV prevention services, including lack of transportation and conflicting priorities. For instance, even when transportation was covered by insurance, women often needed to schedule rides days in advance [31]. The complexity of navigating healthcare processes—such as scheduling appointments for testing, Pap smears, and consultations—added to the challenges faced by women, especially those balancing caregiving responsibilities, housing instability, and employment obligations. This context highlights the broader systemic factors that complicate access to PrEP.

#### 3.1.5. Medical Mistrust

Medical mistrust was identified as a significant barrier in seven studies (*n* = 7) [5,25,26,27,30,34,35]. Many participants cited a historical context of medical experimentation and discrimination against Black individuals in the U.S., including references to the Tuskegee Syphilis Study [25,26,27,35]. This legacy contributes to widespread distrust of the healthcare system. Conversations about medical mistrust frequently intersected with concerns about PrEP side effects [5,27]. Some women expressed skepticism about pharmaceutical companies and their motivations in HIV research, arguing that if a cure for HIV were available, it would not be distributed equitably to the Black population [5]. Overall, medical mistrust significantly impairs women’s willingness to engage with healthcare services.

#### 3.1.6. Low HIV Risk Perception

Low HIV risk perception was discussed in six studies (*n* = 6) [24,26,27,31,32,35]. Many women viewed themselves as being at low risk for HIV due to factors such as being celibate, married, or in a monogamous relationship. D’Angelo et al. (2021) [26] described this preference for monogamy as an “interpersonal” barrier to PrEP uptake, noting that women felt that taking PrEP might signal distrust toward their partners, potentially leading to conflict. Interestingly, participants often recognized a disparity between their own risk perception and that of their community, acknowledging that while they may not view themselves as high-risk, they perceive others in their community to be at significant risk.

#### 3.1.7. Doubts About Pill Adherence

Doubts about pill adherence were noted in five articles (*n* = 5) [5,25,27,32,33]. Women expressed concerns that adhering to a daily regimen of PrEP felt unrealistic given their multiple daily responsibilities. Many women indicated that those at risk for HIV might also be managing challenges such as depression or low self-esteem, which complicates the ability to maintain a consistent routine [25]. Additionally, women already taking hormonal contraceptives viewed PrEP as an added burden. These concerns emphasize the need for alternative PrEP delivery methods that could better accommodate women’s lifestyles.

#### 3.1.8. PrEP Costs

Concerns about the cost of PrEP remain a significant barrier to its accessibility (*n* = 4) [25,26,28,33]. Several women, even those with health insurance, expressed concern about potential out-of-pocket expenses that can make accessing PrEP difficult or impossible.

### 3.2. Recommended Facilitators

While all articles reviewed predominantly focused on barriers to PrEP uptake, nine studies (*n* = 9) identified several potential facilitators [24,25,26,28,30,31,33,34,35].

Table 3 summarizes the key facilitators to PrEP access and uptake, ranked from most to least cited, with the number of studies highlighting each factor.

#### 3.2.1. Increase PrEP Education

Increasing PrEP education was the most frequently recommended facilitator, highlighted in six articles (*n* = 6) [24,26,31,33,34,35]. Participants frequently reported a lack of knowledge about PrEP but expressed a strong desire to learn more when provided with initial information. Common questions centered around PrEP side effects, availability, and administration methods.

To enhance PrEP education, suggested strategies include leveraging online platforms, hosting community sessions in spaces frequented by Black women, and ensuring healthcare providers and social workers receive comprehensive training on PrEP. Misconceptions among healthcare providers can hinder effective recommendations, making it crucial to address these gaps in knowledge. Recommended educational venues encompass social media campaigns, medical clinics, beauty salons, school campuses, and health departments. Additionally, materials should be tailored for women to foster relatability [26].

#### 3.2.2. Empower Black Women as PrEP Advocates

Empowering Black women to serve as advocates and educators for PrEP was discussed in four studies (*n* = 4) [24,30,34,35]. This empowerment is essential in clinical encounters, particularly given the historical context of racism and sexism that affects healthcare interactions. Many Black women reported feeling disempowered due to societal stereotypes and may hesitate to engage with new sexual health information [24,35].

Strategies for empowerment include peer-guided education and sharing testimonials from other Black women who have successfully navigated PrEP [35]. Ensuring the privacy and confidentiality of participants can further enhance their comfort in advocating for PrEP, fostering a supportive environment that encourages open dialogue [24,35].

#### 3.2.3. Fully Cover the Cost of PrEP

Fully covering the cost of PrEP emerged as a significant facilitator in four articles (*n* = 4) [25,26,28,33]. Many participants emphasized the necessity of insurance coverage for PrEP, with few directly inquiring about costs. To reduce financial barriers, recommendations included making PrEP available at no cost in free clinics, thereby increasing access for those who may otherwise be unable to afford it.

#### 3.2.4. Normalize PrEP

The need to normalize PrEP in public discourse and daily life was highlighted in three articles (*n* = 3) [24,26,31]. D’Angelo et al. (2021) [26] emphasized the importance of visibility and community acceptance of PrEP, calling for a “need for increased community consciousness”. Strategies for normalization involve diversifying advertisements to represent cisgender Black women and incorporating PrEP messaging on medical facilities’ websites. Regular and consistent communication about PrEP can help foster greater awareness and acceptance among potential users [26].

#### 3.2.5. Strengthen Trust

Lastly, strengthening trust between Black women and their healthcare providers was noted in three studies (*n* = 3) [5,25,34]. Trust is crucial for PrEP uptake; women who have confidence in their providers are more likely to consider using PrEP, even amidst broader mistrust of the medical system. Healthcare providers must be knowledgeable about PrEP and able to communicate its risks and benefits in a supportive, non-judgmental manner. Additionally, ensuring confidentiality and privacy is essential for establishing and maintaining this trust.

## 4. Discussion

The focus on qualitative studies is meant to summarize data exploring barriers and facilitators to PrEP by Black cisgender women from their own perspectives. This literature review aimed for studies conducted after 2012 because this is the year that the FDA approved PrEP oral pills. The findings demonstrate that there is a scarcity of literature exploring Black women’s perspectives on PrEP. High rates of HIV acquisition amongst Black cisgender women have seemingly only recently become a matter of attention and concern, highlighting the historical neglect of this demographic in the arena of HIV prevention strategies.

### 4.1. Overlapping Barriers

Identified barriers often overlap. For example, three studies described women as being “angry” that their medical providers had never discussed PrEP with them. This lack of PrEP awareness contributed to low perception of HIV risk amongst these women [5,25,34]. They did not perceive themselves to be at risk because their healthcare providers did not seem to perceive them as at risk. The women in the Auerbach et al. (2015) [25] study even expressed concern that they (the women) would be tasked with educating their providers about PrEP. Hirschhorn et al. (2020) found that the women felt that PrEP was being intentionally kept a secret, expressing a sense of medical and governmental mistrust [34].

Similarly, discussions surrounding PrEP side effects often implied medical mistrust. For example, women interviewed in both the Bond et al. (2022) [5] and D’Angelo et al. (2021) [26] studies expressed that they did not trust PrEP medication because they did not trust pharmaceutical companies nor the government agenda surrounding HIV research. One woman in the Bond et al. (2022) [5] study compared the hesitation to take PrEP to how some people hesitate to take vaccines. Put simply, if women do not trust the institution, they will not trust the medicines offered from that institution. Actual side effects of PrEP include a small decrease in bone density among African women and a small decrease in renal function among those taking PrEP [36]. Otherwise, PrEP shows no evidence of an increase in the proportion of adverse events. Furthermore, PrEP is not associated with any adverse pregnancy-related events.

Overall, the concerns about medication side effects are consistent with previous literature that highlighted how medical mistrust reduces medication adherence amongst Black/African American individuals living with HIV [37,38]. This mistrust is rooted in historical experiences of discrimination and exploitation within the United States healthcare system, such as the Tuskegee Syphilis Study [30,34]. Of course, medical mistrust is not only tied to historical exploitation but also exacerbated by gendered power dynamics in healthcare settings, where women’s health concerns are sometimes overlooked or dismissed by male or white healthcare providers [39]. Medical mistrust continues to influence individual health choices, remaining a barrier to accessing effective healthcare interventions.

PrEP stigma was also a significant barrier to PrEP uptake. This finding is consistent with other studies that highlight fear of judgment as a deterrent to HIV prevention efforts [40]. The stigma highlighted in this study is likely deeply influenced by gendered expectations about sexuality. Women reported fears of being labeled as promiscuous or unfaithful by their partners, with several participants expressing fear that introducing PrEP might ignite suspicion of infidelity [5,26,31]. These findings highlight how PrEP stigma, coupled with interpersonal power dynamics, can complicate PrEP uptake. Moreover, these concerns may discourage women from seeking information and education regarding PrEP, further contributing to a cycle of misinformation.

Structural poverty and social determinants of health also play a critical role in PrEP access, as they limit the ability of women to prioritize healthcare amidst issues such as lack of transportation, employment obligations, and need for childcare [24,27]. Concerns about medication costs are intertwined with the issue of structural poverty. Women indicated that the lack of health insurance or out-of-pocket expenses would deter them from seeking PrEP [5,25].

Overall, socio-economic challenges tied to institutional racism—such as limited access to affordable healthcare, lack of HIV education, and insufficient community-based health resources—create significant barriers for Black women in accessing preventive healthcare services like PrEP. Even when Black women are aware of HIV risks, barriers like limited healthcare accessibility, a deep-seated distrust of the medical system due to historical mistreatment, and stigma surrounding HIV and PrEP use prevent them from seeking or adhering to preventive interventions. These barriers are further compounded by the sexual network dynamics, where Black women may be more likely to encounter partners in higher-risk populations due to factors like the sex ratio imbalance, mass incarceration, and higher mortality rates among Black men. These overlapping barriers demonstrate how the systemic inequities that contribute to higher HIV acquisition rates also create complex obstacles in accessing essential HIV prevention measures such as PrEP. While some barriers, such as limited healthcare accessibility and medication stigma, may be shared by women across different racial and ethnic groups, the intersection of these challenges with systemic racism and specific socio-economic factors uniquely exacerbates their impact on Black women. These include historical medical mistreatment, racialized mistrust of healthcare institutions, and the disproportionate exposure to high-risk sexual networks driven by structural inequalities.

### 4.2. Overview of Recommended Facilitators

Additionally, the findings from this study also highlighted several important facilitators to PrEP uptake. These include the importance of PrEP education, social normalization, trust building, empowerment, and financial access. Recommended strategies include embedding PrEP education in community settings like clinics, schools, beauty salons, and online platforms, and ensuring that materials are culturally relevant and visibly representative of Black women [24,33,35]. This is consistent with Johnson et al.’s (2021) study that found beauty salons have been identified as a trusted space for health promotion among Black women [41]. Furthermore, the normalization of PrEP through targeted media campaigns and community-based outreach can help reduce stigma and make PrEP a common, trusted option in sexual health care. According to D’Angelo et al. (2021) [26], frequent and positive messages about PrEP can elevate community consciousness, helping Black women to see PrEP as a standard preventive measure.

Strengthening the trust between Black women and healthcare providers is a key facilitator, as positive patient–provider relationships increase openness to PrEP amidst the broader barrier of medical mistrust. Healthcare professionals must be trained to deliver nonjudgemental and supportive care [25,34]. Moreover, empowering Black women as PrEP advocates and peer educators can address both informational and social barriers, offering role models who normalize PrEP through relatable testimonials and shared lived experiences. Finally, ensuring the affordability of PrEP by advocating for insurance coverage and making it available in free or reduced-cost clinics can mitigate the financial barriers that deter PrEP use, especially in underserved communities [26,33].

Both institutional and interpersonal racism impact how Black/African American people experience healthcare [42]. The implications of this study underscore the urgent need for tailored HIV prevention strategies that prioritize the perspectives of Black cisgender women, particularly in the context of institutional and interpersonal racism. Notably, in the Bond et al. (2022) [5] article, women suggested that it is important to have the direction of a doctor when introducing the subject of PrEP to a romantic partner. The same article describes medical mistrust as a barrier to PrEP uptake, suggesting that establishing a good relationship between provider and patient is a crucial first step in overcoming medical mistrust. Women who trust their healthcare providers are more likely to listen to their providers. Thus, it is important for healthcare providers to be reflexive about internalized racist biases and stereotypes. Furthermore, community-based education and peer-led advocacy can empower Black women, enabling them to make informed health decisions and potentially serve as advocates within their communities. It is also important to integrate PrEP into insurance coverage and free clinic programs, dismantling financial barriers rooted in systemic inequities.

### 4.3. Limitations

This literature review has several limitations. Despite our efforts to be exhaustive, it is possible that some search terms were unintentionally omitted, which could have yielded additional relevant results. Additionally, there are databases that we did not explore, and this review did not include ongoing research studies, implementation studies, or dissertations. We also excluded studies focusing on specific populations, such as incarcerated women, substance users, and adolescents, which may have enriched our data. For example, research on incarcerated women might have shed light on how institutional settings impact access to and perceptions of PrEP. Similarly, studies on substance users could have illuminated the intersections of addiction, stigma, and preventive health behaviors, while research on adolescents might have contributed valuable perspectives on early interventions, health education, and parental or societal influences on PrEP decision-making. By not including these populations, we may have missed critical nuances and variations in the challenges and opportunities related to PrEP uptake.

### 4.4. Implications for Public Health Research

This review highlights several key areas for future research to address barriers and facilitators to PrEP uptake among Black cisgender women. The scarcity of studies exploring their perspectives underscores the need for additional qualitative and mixed-methods research capturing the lived experiences of Black cisgender women across diverse geographic, socioeconomic, and cultural contexts. Such research can provide a comprehensive understanding of the factors influencing PrEP uptake and inform targeted interventions.

Given the overlap between barriers like medical mistrust, stigma, and low awareness, future studies should investigate how these factors interact and cumulatively impact PrEP utilization. Longitudinal research could offer valuable insights into how these barriers evolve over time and the long-term effects of targeted interventions. Moreover, systemic solutions—such as expanded insurance coverage, free clinic programs, and policies addressing social determinants of health—warrant further evaluation to determine their effectiveness in reducing financial and logistical barriers to PrEP access.

Provider-focused research is equally critical, particularly to assess the impact of training programs that address internalized biases, enhance cultural competence, and foster nonjudgmental patient-provider communication. An intersectional approach is essential to understanding how overlapping identities, such as gender, race, socioeconomic status, and sexual orientation, shape Black women’s experiences with PrEP. This nuanced understanding can help identify unique challenges faced by subgroups within this population and guide the development of tailored interventions.

Additionally, the findings of this study emphasize the value of community-based participatory research (CBPR) approaches, which actively involve Black women in shaping research priorities, intervention designs, and policy recommendations. This participatory model ensures culturally relevant interventions that directly address the lived realities of this population. Innovative strategies for integrating PrEP education and services into trusted community spaces—such as faith-based organizations—should also be explored to enhance accessibility and reduce stigma.

For future research, incorporating a deeper gender perspective is important for understanding how intersections of gender, race, and class may amplify barriers to accessing PrEP. Research should also investigate the impact of stigma, including how perceptions of promiscuity can influence women’s decisions regarding PrEP uptake. Further studies comparing Black cisgender women’s experiences with other women’s groups, such as Latina or White women, could identify specific disparities and inform more targeted interventions. In addition, while this study centers on cisgender women, future research should consider how transgender women’s experiences with PrEP differ from cisgender women’s experiences in an effort to understand how prevention strategies can be made more inclusive and effective for all women.

While this study focused on oral PrEP, future research should examine the challenges and opportunities surrounding long-acting PrEP, particularly given its potential to address adherence issues. Although long-acting PrEP should have emerged in our literature search, no studies on this intervention were found among the 13 articles meeting inclusion criteria. This omission highlights the limited scope of existing research and underscores the need for further investigation into this emerging area of HIV prevention [43].

By addressing these gaps, public health research can develop holistic, equity-driven strategies to enhance PrEP uptake and reduce HIV disparities among Black cisgender women, ultimately contributing to broader efforts to achieve health equity and improve HIV prevention outcomes.

## 5. Conclusions

In conclusion, this study explored critical barriers and facilitators to PrEP uptake among Black cisgender women by centering their own perspectives. The results of this study suggest an urgent need for more research on the barriers and facilitators to PrEP uptake among Black cisgender women. This group is disproportionately affected by HIV yet historically neglected in prevention strategies. The findings reveal overlapping barriers, including medical mistrust, stigma, and structural poverty, which are compounded by systemic inequities and healthcare providers’ failure to discuss PrEP. Distrust in institutions, stemming from historical injustices like the Tuskegee Syphilis Study, further deters PrEP uptake, while stigma and interpersonal dynamics complicate its introduction in relationships.

Facilitators such as community-based PrEP education, peer advocacy, and improved affordability and access can help address these barriers. Building trust through culturally competent healthcare and addressing systemic racism are critical steps toward reducing disparities. Though existing studies are limited in scope, future research should adopt intersectional approaches, explore systemic solutions, and address gaps in knowledge to inform equity-driven interventions that enhance PrEP uptake among Black cisgender women.

## Figures and Tables

**Figure 1 healthcare-13-00086-f001:**
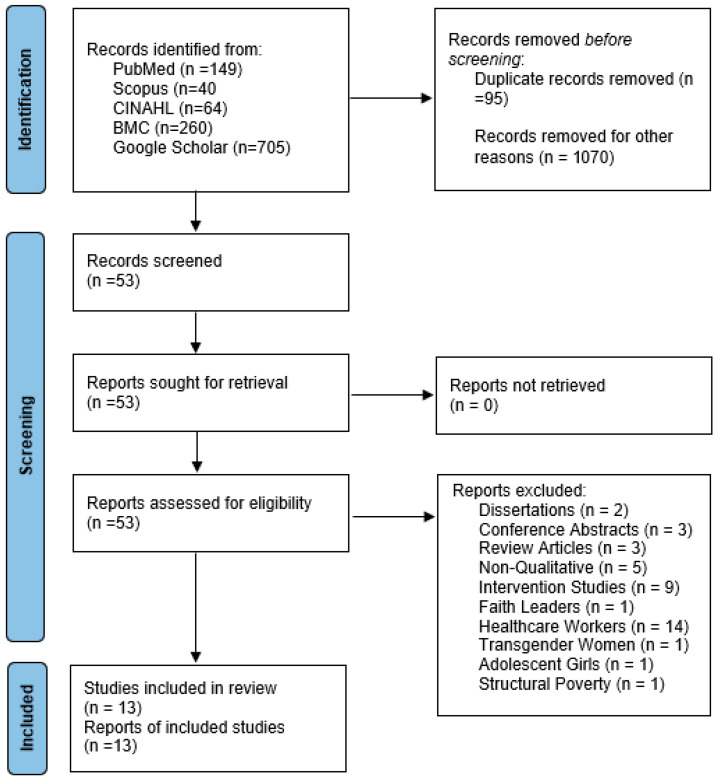
Systematic Review Flow Diagram Following PRISMA Guidelines Source: Developed by authors.

**Table 1 healthcare-13-00086-t001:** Risk of bias assessment using CASP checklist.

Study	Risk of Bias Assessment	Summary of Risk of Bias
Bond et al. (2022) [5]	Low	The study’s findings are specific to urban settings, particularly New York City. This may limit the generalizability of the results to rural areas or different geographic contexts. This limitation should be noted when interpreting the findings, as the experiences of Black women in urban environments may differ significantly from those in rural settings.
Arnold et al. (2022) [24]	Low	The study’s findings are specific to the context of Jackson, Mississippi. This may limit the generalizability of the results to other geographic areas. This limitation should be noted when interpreting the findings, as the experiences of Black women in Mississippi may differ from those in urban or rural settings elsewhere.
Auerbach et al. (2015) [25]	Low	The study’s findings are specific to urban settings, particularly in cities like New York, Dallas, and Atlanta. This may limit the generalizability of the results to rural areas or different geographic contexts. This limitation should be noted when interpreting the findings, as the experiences of women in urban environments may differ significantly from those in rural settings.
D’Angelo et al. (2021) [26]	Low	The study’s findings are specific to urban settings, particularly in New York City. This may limit the generalizability of the results to rural areas or different geographic contexts. This limitation should be noted when interpreting the findings, as the experiences of Black women in urban environments may differ significantly from those in rural settings.
Nydegger et al. (2021) [27]	Low	The study’s findings are specific to the context of Milwaukee, Wisconsin. This may limit the generalizability of the results to other geographic areas or different types of institutions. This limitation should be noted when interpreting the findings, as the experiences of Black women in Milwaukee may differ from those in urban or rural settings elsewhere.
Park et al. (2019) [28]	Moderate	The study’s findings are specific to women recruited from an urban sexual health clinic in the Bronx, New York. While the insights into facilitators and barriers to PrEP uptake are valuable, the results may not be generalizable to women in other geographic areas or healthcare settings without similar resources. This limitation should be considered when interpreting the findings, as the experiences of women in different contexts may vary significantly.
Pyra et al. (2022) [29]	Low	The study’s findings are specific to Chicago, Illinois. This may limit the generalizability of the results to other geographic areas or different types of institutions. This limitation should be noted when interpreting the findings, as the experiences of Black women in Chicago may differ from those in urban or rural settings elsewhere.
Randolph et al. (2020) [30]	Moderate	The study focuses on Black women living in low-income housing in the southern United States, specifically Chapel Hill, North Carolina. While the findings provide valuable insights into barriers faced by this population, the results may not be generalizable to Black women in other geographic areas or socioeconomic contexts. This limitation should be considered when interpreting the findings, as the experiences of women in different settings may vary significantly.
Troutman et al. (2021) [31]	Low	The study’s findings are specific to the Southeastern United States. While the findings provide valuable insights into this specific population, the results may not be generalizable to Black women in other regions. This limitation should be considered when interpreting the findings, as the experiences of women in various contexts may differ significantly.
Chandler et al. (2020) [32]	Moderate	The study’s findings are specific to the context of an HBCU, which may limit the generalizability of the results to other geographic areas or different types of institutions. This limitation should be noted when interpreting the findings, as the experiences of Black women in HBCUs may differ from those in predominantly White institutions or rural settings.
Collier et al. (2017) [33]	Low	The study’s findings are specific to urban settings, specifically the Bronx, New York. This may limit the generalizability of the results to other geographic areas or different types of institutions. This limitation should be noted when interpreting the findings, as the experiences of Black women and Latinas in the Bronx may differ from those in other urban or rural settings.
Hirschhorn et al. (2020) [34]	Low	The study’s findings are specific Chicago, Illinois. This may limit the generalizability of the results to other geographic areas or different types of institutions. This limitation should be noted when interpreting the findings, as the experiences of Black women in Chicago may differ from those in urban or rural settings elsewhere.
Willie et al. (2022) [35]	Moderate	The study’s findings are specific to the city of Jackson Mississippi. This may limit the generalizability of the results to other geographic areas or different types of institutions. This limitation should be noted when interpreting the findings, as the experiences of Black women in Mississippi may differ from those in urban or rural settings elsewhere.

**Table 2 healthcare-13-00086-t002:** Barriers to PrEP access and uptake identified across articles.

Barrier	Number of Studies
Low Levels of PrEP Knowledge	12
PrEP Stigma	11
Concerns about PrEP Side Effects	9
Structural Poverty and Social Determinants	7
Medical Mistrust	7
Low HIV Risk Perception	6
Doubts about Pill Adherence	5
PrEP Costs	5

**Table 3 healthcare-13-00086-t003:** Facilitators to PrEP access and uptake identified across articles.

Barrier	Number of Studies
Increase PrEP Education	6
Empower Black Women as PrEP advocates	4
Fully Cover the Cost of PrEP	4
Normalize PrEP	3
Strengthen Trust	3

## Data Availability

Data sharing is not applicable. No new data were created or analyzed in this study.

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
