# Peer review of "Identifying Access Barriers to PrEP Among Cisgender Black/African American Women in the United States: A Systematic Review of the Literature"

_healthcare, 2025, doi:10.3390/healthcare13010086_

Round 1

Reviewer 1 Report

Comments and Suggestions for Authors

Introduction

1.Line 69 states, “Since then, PrEP has contributed to an 18% reduction in HIV infections from 2015 to 2019.” It is advised to cite relevant references.

2.The text could be more concise. The last three paragraphs discuss the rationale, objective, and aim of this review. It is suggested to merge them.

3.Line 56 already provides the abbreviation for pre-exposure prophylaxis (PrEP); however, the abbreviation is repeated in line 92, as well as in other parts of the text. After the first appearance of the abbreviation, it should be used directly in subsequent mentions.

Materials and Methods

1.Long-acting PrEP has been approved in the U.S. Given that oral PrEP generally has limited effectiveness for women, barriers to the use of long-acting PrEP for women should be a greater focus. However, the search terms in this study do not reflect the relevant content of long-acting PrEP.

Results

2.In Figure 1, for "records removed for other reasons," please specify the reasons.

3.In Figure 1, what does "records excluded**" mean? There is no related explanation provided.

4.Table 1 lists the authors, year, title, and study location of 13 articles. Since the title is not the main focus, it is recommended to remove it. The "year" is not specific enough; it should be clarified as the time of the study or publication. 

5.Tables 2 and 3 could be combined into one table.

6.When discussing barriers, has the influence of cultural and social contexts on Black cisgender women been sufficiently considered?

Discussion

1.The section “Implications for Future Research” should include content related to long-acting PrEP.

2.The 13 studies included in this research are primarily focused on major cities in the East and West, such as New York and Chicago, which may not represent the entire United States. This limitation should be noted, and it is also suggested that the paper's title be made more specific, changing it to “Certain Regions of the United States” might be better.

Minor Revisions

1.Line 197: “While all fourteen studies explored barriers to PrEP.” The final included studies count is only 13. Where does the term "fourteen" come from?

Author Response

Please see the attachment for Response to Reviewer 1

Reviewer 2 Report

Comments and Suggestions for Authors

Thank you for the opportunity to review the manuscript titled "Identifying Access Barriers to PrEP among Cisgender Black/African American Women in the United States: A Systematic Review of the Literature." This study addresses critical health disparities related to PrEP access among Black women in the U.S., which is highly relevant to controlling the HIV epidemic in this population. However, the current version requires substantial revisions to meet the necessary standards for publication. Below are my detailed comments:

Introduction

The manuscript provides insufficient explanation for why the focus is specifically on Black women. I recommend that the authors emphasize this choice, particularly in terms of the unique challenges faced by this group in accessing healthcare and HIV prevention methods.

The introduction is overly lengthy for a systematic review. I suggest the authors condense this section, or alternatively, move some of the content to the discussion section, as much of it relates directly to the study’s findings.

The authors need to clearly justify why only qualitative studies were included in the review, rather than quantitative studies. This will likely be a point of curiosity for readers. Generally speaking, quantitative studies are broader in scope and can identify more factors, whereas qualitative studies provide deeper insights into the underlying reasons behind these factors. Given the authors’ focus on “barriers,” it might be beneficial to discuss both qualitative and quantitative research to provide a more comprehensive understanding of the issue.

Results

I recommend that the authors consider adding a column in Table 2 for keywords that will allow readers to quickly identify the key "barriers."

Additionally, I suggest that the authors remove the term "articles" from the subheadings. Instead, consider using an Appendix or a table to list the number of studies related to each "barrier." This would improve clarity and presentation.

Discussion

The purpose of a systematic review is to synthesize existing findings and propose directions for future improvement. The public health significance of these findings should be emphasized more clearly. In this section, I recommend that the authors reorganize the discussion by using subheadings to address each identified "barrier," offering an in-depth discussion of the underlying causes and suggesting strategies to address them. This will help strengthen the manuscript's public health relevance.

It is crucial for the authors to highlight whether these "barriers" are common to all U.S. women, regardless of race, or if they are uniquely faced by Black women. This distinction is vital to the paper's focus on “health inequities.” The authors should clarify whether the barriers faced by Black women are part of a broader issue or a specific challenge for this group.

Furthermore, I am curious whether Black men in the U.S. face similar "barriers" as Black women in accessing PrEP. This could be incorporated into the discussion to offer a more comprehensive insight into the broader challenges facing Black Americans in accessing HIV prevention.

In conclusion, while this manuscript tackles an important and timely issue, the authors must refine the manuscript by addressing the above points. These revisions would strengthen the paper’s clarity, scientific rigor, and its public health relevance.

Thank you again for the opportunity to review this manuscript. I look forward to the revised version and hope my comments will help enhance the quality of the paper.

Author Response

Please see attached Responses to Reviewer #2

Reviewer 3 Report

Comments and Suggestions for Authors

Thank you very much for sending the article on “Identifying Access Barriers to PrEP Among Cisgender Black/African American Women in the United States: A Systematic Literature Review.” I found the manuscript highly engaging, so congratulations on your work, and I look forward to seeing its progression.

The article stands out for its clear objectives, methodological rigor, relevance, and timeliness. Additionally, it provides a well-balanced discussion connecting findings to structural contexts and identifying practical solutions, such as community-based education and the empowerment of advocates. Below, I offer some recommendations that may help further improve the article:

Abstract: The abstract should be more concise, clearly emphasizing the results and conclusions. Including a brief mention of the methodology (e.g., PRISMA and CASP) would also highlight the rigor of the approach.

The introduction presents valuable context but could benefit from a clearer structure. I suggest reorganizing it to start with a general overview of HIV disparities in the U.S., followed by a specific focus on cisgender Black women and the barriers they face.

It could delve deeper into why this particular group (cisgender Black women) has been historically underserved in PrEP research, emphasizing how their vulnerability is influenced by unique structural and cultural factors.

While the study’s objective is well-defined, it would be helpful to link it more directly to the initial context. For instance, you might include how the study addresses specific gaps in existing literature regarding structural and social barriers.

Consider revising and including key references to support general claims, such as data on HIV rates among cisgender Black women or studies highlighting how structural racism impacts their access to healthcare.

It may also be valuable to briefly mention how an intersectional analysis of gender and race will be integral to the study’s approach.

A smoother transition into the methodology section would help readers connect the presented context with the systematic review.

Methodology: While exclusion criteria are mentioned, it would be helpful to provide a more detailed justification for excluding populations such as incarcerated women, adolescents, and transgender women, clarifying how these omissions could affect the generalizability of findings.

PRISMA Flow Diagram (Figure 1): Add a clear title at the top (e.g., “PRISMA Flow Diagram”), and include the source at the bottom, such as “Source: Developed by the authors.”

Consistency in Definitions and Terminology: Review this section to ensure consistent usage of key terms like “systemic racism” and “medical mistrust.” Unify terminology when referring to participants (e.g., “cisgender Black women” or “African American women”) to avoid confusion.

Thematic Connections: Strengthen the relationships between key themes, such as the link between low PrEP awareness and provider biases, to show how these factors interact and reinforce each other.

Include case studies or concrete examples of successful programs that have implemented facilitators, such as community-based education or empowerment strategies, to illustrate how they could be applied in other contexts.

Data Extraction Process:Provide details on how discrepancies between reviewers were managed during the selection and data extraction process and whether any measures were taken to assess inter-rater reliability.

Discussion: Explicitly link facilitators to structural barriers. For example, explore how strategies like trust-building and community education can mitigate the effects of institutional racism and structural poverty.

Study Limitations: Expand on how limitations, such as the focus on urban settings, may have influenced the findings. It would also be helpful to suggest strategies to overcome these limitations in future research.

Recommendations for the Future: Review this section and propose concrete methodologies, such as longitudinal studies or intersectional designs, to evaluate how barriers evolve over time and how specific policies can address the disparities identified. Additionally, clarify how subsequent parts of the study might address these issues.

Gender Perspective: Incorporate a deeper gender perspective by analyzing how power dynamics, stigma, and gender roles influence barriers and facilitators. Reflect on differences between cisgender and transgender women’s experiences to design more inclusive prevention strategies, which I understand may be part of the project’s continuation.

Writing and Style: Simplify complex sentences, especially in the introduction and discussion, to enhance readability and appeal to a broader audience. Avoid redundancies and ensure logical flow between sections.

References: Ensure that updated and seminal works are included and verify that in-text citations are consistent with the final reference list.

For future research, I recommend incorporating a more inclusive and profound gender perspective that analyzes how intersections of gender, race, and class amplify barriers to accessing PrEP. Investigate the impact of stigma related to perceptions of promiscuity, include comparisons with other groups of women to identify specific disparities, propose policies and interventions tailored to women’s needs, and reflect on how cisgender women’s experiences relate to those of transgender women to design more equitable and effective prevention strategies.

I hope these suggestions assist in refining the manuscript and advancing your research project. Congratulations, and I look forward to seeing your article published soon

Author Response

Please see attached Responses to Reviewer #3

Round 2

Reviewer 2 Report

Comments and Suggestions for Authors

The authors addressed all my comments and I recommend acceptance.

Author Response

Thank you for sharing the reviewer’s feedback on our manuscript. We are deeply grateful for the time and effort the reviewer dedicated to evaluating our work and for their positive comments regarding the quality of the research and its presentation.

We are pleased that the reviewer found the manuscript clear, well-structured, and supported by the results. We appreciate their recommendation for acceptance and look forward to the next steps in the process.

Thank you once again for this opportunity.